# Willingness to Test for Human Immunodeficiency Virus (HIV) Infection among First-Year Students of a Public University in the Volta Region of Ghana

Mispa Tepe-Mensah [1], Joseph Osarfo [1], Evans Kofi Agbeno [2] and Gifty Dufie Ampofo [1,*]

1 Department of Community Medicine, School of Medicine, University of Health and Allied Sciences, Hohoe PMB 31, Ghana
2 Department of Obstetrics and Gynaecology, School of Medical Sciences, University of Cape Coast, Cape Coast P.O. Box 5007, Ghana
* Correspondence: gampofo@uhas.edu.gh

**Abstract:** Voluntary counselling and testing (VCT) is key in HIV prevention. Young people aged 15–24 years carry a significant burden of new infections globally, but VCT uptake is low in this population. The study assessed university freshmen's willingness to test for HIV now, among others, in a cross-sectional study as university campuses are places of risky sexual behaviour. Structured questionnaires were used to collect data on age, sex, marital status, HIV/AIDS knowledge, previous history of testing, willingness to test now, and others. Summary statistics were reported while chi-square and logistic regression methods were used to assess the association between dependent and independent variables with $p$-values < 0.05 held significant. About 90% (374/412) of respondents had good HIV/AIDS knowledge based on criteria defined by the study, but only 23.3% (96/412) had ever tested and 66.3% (266/401) were willing to test now for HIV. Respondents' sex, previous sexual intercourse, and whether respondents' educational support was from parents or non-parents influenced willingness to test for HIV now. The study highlights what appears to be personal beliefs that can potentially hinder HIV testing and control efforts. Relevant stakeholders must address these gaps to improve testing. Further qualitative investigation will improve understanding of the dynamics informing willingness to test for HIV among young people generally.

**Keywords:** HIV; willingness to test; VCT uptake; knowledge; first-year university students; Ghana

## 1. Introduction

Human immunodeficiency virus (HIV) infection and acquired immune deficiency syndrome (AIDS) remain issues of significant public health importance with a high burden of morbidity and mortality [1,2]. Globally, over 39 million AIDS-related deaths had occurred by 2019 and there are close to 2 million new infections every year [1,2]. In 2019 and 2020, AIDS-related deaths averaged 690,000 per year and though appreciable, this was a 39% reduction compared to estimates for 2010 figures [3,4]. An estimated 37.7 million people were living with HIV in 2020 and 84% knew their status compared to 81% in 2019 [3,4]. Voluntary counselling and testing (VCT) to know one's status is a key prevention intervention which also promotes early treatment and care [5]. However, getting people to test of their own free will remains a major challenge [6]. This challenge is even more pronounced in young people aged 15–24 years as the proportion of people living with HIV who know their status is believed to be significantly lower in this population although they harbour about a third of all new infections globally [7,8].

Close to 50% of young people living with HIV, including those aged 15–24 years, are not diagnosed and linked to treatment to achieve viral suppression [7,9]. This situation stems from a low uptake of HIV testing services among them and threatens prevention and control efforts. The Joint United Nations Programme for HIV/AIDS' target of having 95%

of people living with HIV know their status by 2030 [3] is likely to be threatened by such low uptake or willingness to test.

Sub-Saharan Africa is home to two-thirds of global HIV infections with about 26 million people living with HIV/AIDS and 70% of AIDS-related deaths [10]. About 33% of new infections in this region in 2019 [3,10] occurred in the 15–24 age group in 2019. Low uptake of HIV testing has also been observed in this region and only about 35% of Ethiopian youth were found to have ever tested for HIV [11]. In Ghana and Kenya, 80–95% of young people aged 15–24 years, including university students, knew where to access HIV testing services but only about 45% had previously tested for HIV [12–14]. This suggests that access to testing services may not directly translate into uptake [12,13] and reinforces the observation of low willingness to test for HIV in this age group [5]. In contrast, other studies have also reported high uptake of HIV testing among university students [13,15,16] and the dynamics contributing to the variations need to be understood.

Higher educational levels, moderate-to-comprehensive knowledge of HIV/AIDS, and risky sexual behaviour are positively associated with voluntary testing uptake [11], while stigma, perceived low risk of HIV infection, perceived lack of confidentiality on the part of health workers, fear of test outcomes, and the perceived psychological burden of living with HIV can hamper uptake of VCT [17–19]. Being male has been noted as both a barrier and a facilitator to VCT [11,12].

In Ghana, 28% of the almost 19,000 new infections in 2020 occurred in the 15–24 years age group [20]. The HIV prevalence in this age group was 1.5% over 2017 and 2018 but declined to 0.7% in 2020, translating to about 42,000 young people living with HIV [20]. With the reported low uptake of VCT in this age group [7,9], the prevalence may be higher. It is possible many more young people are living with the infection unknowingly because they have been unwilling to test. Thus, it is essential to assess willingness to test for HIV and the factors associated in this population to help contribute to knowledge on this subject and suggest evidence-based recommendations for action. Useful insights to guide policy makers in promoting HIV counselling and testing among the youth with the hope of early diagnosis and subsequent linkage to treatment will be attained.

Within this context, it is appropriate studying this phenomenon using first-year university students as a proxy for this age group in question. It has been noted that first-year university students have relatively adequate knowledge on HIV/AIDS but their new-found freedom as university students, independent of direct parental supervision, may lead them to initiate or upscale sexual activity [21]. First-year students at a public university in Ghana were surveyed to ascertain their willingness to test for HIV and the factors influencing this decision. The proportion who had tested for HIV before and the reasons motivating the testing were also assessed. The study was guided by the Health Belief Model (HBM) [22]. This theoretical framework explains behaviour towards health promotion and disease prevention. An individual's personal beliefs or knowledge/perceptions about a disease influences behaviour towards available options to mitigate it [23,24]. Although the HBM is reported to have a relatively limited predictive value compared to other health behaviour models, it has been frequently utilized in studies looking at health services uptake including HIV services [25,26].

## 2. Materials and Methods

### 2.1. Study Design, Population and Setting

A cross-sectional design was employed to conduct a survey among first-year undergraduate students at the Ho campus of the University of Health and Allied Sciences in the Volta Region of Ghana from 1 June 2021 to 31 July 2021. The institution is the only public university in the country dedicated solely to the training of health professionals including doctors, nurses/midwives, pharmacists, and other allied health groups. At the Ho campus, the university operates six schools/faculties: School of Basic and Biomedical Sciences (SBBS), School of Medicine (SOM), School of Nursing and Midwifery (SONAM), School of Allied Health Sciences (SAHS), School of Pharmacy (SOP), and School of Sports & Exercise

Medicine (SOSEM). At the time of study conduct, the total population of undergraduate first-year students in the institution was 836. The HIV prevalence rate in the Volta Region was 1.6% in 2020 and young people 15–24 years numbered 329,617, accounting for nearly 20% of the region's almost 1.7 million population [20,27].

### 2.2. Sample Size Determination

The sample size was estimated using the Cochran formula $n = (z^2pq)/e^2$, where $n$ is the sample size, z is the reliability coefficient at 95% confidence interval, i.e., 1.96, p is the estimated proportion of students willing to test for HIV, q is (1-p), and e is the desired precision level of 5%. An earlier study among university students in Ghana reported 62.7% prevalence of willingness to test for HIV [12]. However, this study included second and third-year students in its sample and was not deemed appropriate for estimating the sample size in the present study. Furthermore, the reported prevalence was deemed out of date considering that the study was conducted about a decade ago. A 'p' of 50% was thus assumed in calculating the sample size and this gave an initial estimate of 384. This was then adjusted upwards by 5% for non-response and a minimum sample size of 403 was arrived at.

### 2.3. Study Procedures and Data Collection

Participants were from all the six schools of the university at the Ho campus and were included if they were aged 18–24 years and willing to participate. Students pursuing sandwich or top-up programmes were excluded as they were not in school at the time of the study. A stratified sampling technique was used in participant selection. The number of participants allocated to each school was calculated as a function of the total number of first-year students in that school, the total number of first-year students in the university as of 1 June 2021, and the calculated sample size (see Table 1). Using the class list for the first-year students in each school as a sampling frame, participants were selected randomly and approached to participate in the study. Where a selected participant declined participation, the selection methods were repeated until the allocated numbers were obtained. Data collection was completed in one school before moving to the next until all schools were surveyed within the data collection period.

**Table 1.** Stratum/School and number of participants allocated.

| Stratum/School | * Number of First-Year Students | Number of Study Participants Allocated |
|---|---|---|
| SOM | 134 | 65 |
| SONAM | 297 | 143 |
| SBBS | 100 | 48 |
| SOP | 54 | 26 |
| SAHS | 238 | 115 |
| SESEM | 13 | 6 |
| Total | 836 | 403 |

* The number of first-year students in each school was sourced from the Academic Affairs Directorate of the University.

After giving written informed consent, self-administered structured questionnaires (see supplementary file S1), developed de novo by the investigators through literature review, were used to collect data on participants' sociodemographic characteristics including age, sex and marital status, knowledge of HIV/AIDS, whether they have ever tested for HIV, willingness to test for HIV/AIDS now, and factors influencing their decision. The dependent variable, willingness to test for HIV now, had a binary response YES or NO.

The level of knowledge of HIV/AIDS was derived from scores in response to seven questions regarding the causative organism, who can get infected, mode of transmission, signs and symptoms of the disease, making a definitive diagnosis, treatment, and prevention. For questions requiring only one correct choice, a correct response earned 1 point. For questions with multiple correct responses, a maximum of 2 points was assigned if at least 2 correct choices were selected and 0 if only one correct response or none was selected. No scores were assigned if the "I don't know" option was chosen. Thus, knowledge scores ranged from 0 to 10. A total score of ≤6 was considered as 'poor knowledge' while a total score of ≥7 was considered 'good knowledge'. The cut-off, a step above 50% of the total score, was chosen to reflect a desired need for participants to have adequate knowledge of HIV/AIDS as this is beneficial in prevention efforts. Similar methods for categorizing knowledge levels pertaining to HIV/AIDS and COVID-19 have been used previously [12,28–30]. The questionnaire was pretested among ten second-year physician assistantship students and appropriate changes were made to its structure before it was used.

### 2.4. Data Management and Analysis

Data was double entered into Microsoft Excel 2016 (Microsoft Corporation, Redmond, WA, USA) for cleaning and imported into Stata version 13 (College Station, TX, USA) for analysis. Descriptive statistics were performed, and results were reported as frequencies, percentages, and means. The effect of exposure variables on the outcome variable "willingness to test for HIV now" was explored first in a bivariate logistic regression analysis. Multivariate regression was done in a stepwise backward elimination process with a threshold *p*-value of 0.1. The independent variables studied were participant's sex, age, whether their hall of residence is privately owned or university owned, whether participants' education is supported by parents or other guardians, the marital status of participants' parents, participants' usual residence outside campus, whether participants have had sex or tested for HIV before, the school the participants belonged to in the university, and their ethnicity. A *p*-value ≤0.05 was considered statistically significant in the final model. Participants' religion and marital status were not included in the multivariate regression because they were markedly homogeneous in distribution.

### 2.5. Ethical Considerations

Ethical approval for the study was obtained from the University of Health and Allied Sciences Research Ethics Committee (Protocol Identification Number: UHAS-REC A.12 [150] 20-21). Written informed consent was obtained from all participants and anonymity was assured by using study identification numbers rather than names. Study participants could withdraw at any time from the study if they so wished without any consequences. The study involved only gathering information by means of questionnaires with no health risk to the participants. Participants were not given any monetary or material benefits.

## 3. Results

### 3.1. Participants' Sociodemographic Characteristics

Four hundred and twelve (412) first-year students participated in the survey. Table 2 below summarizes the participants' sociodemographic characteristics. The mean age (SD) was 20 years (1.7 years). Participants aged 18–20 years accounted for 67.2%. Females made up 50.9% of the study population. The participants were nearly all Christians (94.1%), and 75.1% had urban settings as their usual residence outside campus.

**Table 2.** Sociodemographic characteristics of participants.

| Socio-Demographic Characteristics | Frequency | Percentage (%) |
|---|---|---|
| (years) (*n* = 408) | | |
| ≤20 | 274 | 67.2 |
| >20 | 134 | 32.8 |
| Sex (*n* = 409) | | |
| Male | 201 | 49.1 |
| Female | 208 | 50.9 |
| Religion (*n* = 407) | | |
| Christianity | 383 | 94.1 |
| Islam | 20 | 4.9 |
| [a] Others | 4 | 1.0 |
| Marital status (*n* = 409) | | |
| Married | 3 | 0.7 |
| Not married | 406 | 99.3 |
| School (*n* = 411) | | |
| SOM | 70 | 17.0 |
| SONAM | 148 | 36.0 |
| SAHS | 115 | 28.0 |
| SOP | 25 | 6.1 |
| SOSEM | 7 | 1.7 |
| SBBS | 46 | 11.2 |
| Hall of residence (*n* = 409) | | |
| University owned | 341 | 83.4 |
| Privately owned | 68 | 16.6 |
| Educational support (*n* = 394) | | |
| Parents | 348 | 88.3 |
| Guardian | 46 | 11.6 |
| Ethnicity (*n* = 403) | | |
| Ewe | 140 | 34.7 |
| Akan | 186 | 46.2 |
| [b] Others | 79 | 19.1 |
| Marital status of parents (*n* = 409) | | |
| Married | 284 | 69.4 |
| Single/Divorced/Widowed | 125 | 30.6 |
| Place of usual residence outside campus (*n* = 409) | | |
| Urban | 307 | 75.1 |
| Rural | 102 | 24.9 |

[a] Includes one atheist and three devotees of African traditional religion; [b] includes tribes such as Ga-Dangbe, Mole-Dagbani, Kusaasi, Biakpakpaam, Krobo, Frafra, Guan and Hausa, and others.

*3.2. Willingness to Get Tested for HIV Now*

Of the 401 participants who answered the questions, 66.3% were willing to test now, although 68.3% claimed they had never had sex before. Curiosity was the most common reason given for willingness to test now (61.7%). Other reasons were "getting married soon" (8.3%), "I am at risk" (6.0%), and "history of rape" (1.9%). About a fifth (22.2%) gave no reason at all for their willingness to test for HIV now.

Majority of those unwilling to test now gave no reason at all (52.0%). However, some reasons given for their unwillingness to test now included the following: "I can never get infected" (23.7%), "I don't care" (8.8%), and "there is no cure so it makes no difference if I test or not" (5.2%).

*3.3. Ever Tested for HIV*

Only 23.3% of all the respondents had ever tested for HIV. Of these, 44.8% had tested within the year preceding data collection and the rest more than a year prior to the study. Females constituted 53.7%. Reasons giving for taking the test included "Health worker request" (53.2%), "being in a high-risk group of sexually active persons/rape victims/drug

users" (31.6%), fulfilling requirements for marriage, or completing documentation for educational scholarships (22.8%).

Of the 316 respondents who had never tested for HIV before, not being sexually active (33.2%) was the commonest reason given. Indifference towards knowing their status, fear of positive results, stigma, and a perceived lack of confidentiality on the part of health workers accounted for 47.5% of the respondents, and 1.6% thought there is no definitive cure for HIV/AIDS and hence testing to know one's status was not necessary. Other reasons were that they had not had the opportunity to test, did not know where to get tested, did not have HIV, did not share sharps with anyone, or simply did not have any reason to take the test.

### 3.4. Participants' Knowledge Level of HIV/AIDS

Out of the 412 study participants, 90.8% had good knowledge of HIV/AIDS while 9.8% had poor knowledge. Television (59.4%), radio (32.4%), friends (30.2%), and school curricula (41.8%) were the most important sources of HIV/AIDS knowledge among the 404 study participants who responded to this multiple response question.

### 3.5. Factors Associated with Willingness to Test for HIV Now

In the bivariate logistic regression analysis (see Table 3), males were 42% less willing to test for HIV now compared to females (OR 0.58 95%CI: 0.38–0.88; $p = 0.011$). Respondents whose education was supported by their parents were at least twice as willing to test for HIV now compared to those whose education was supported by guardians other than parents (OR 2.36, 95%CI: 1.25–4.45; $p = 0.008$). Participants who had had sexual intercourse before were almost twice as willing to get tested now compared to those who had never had sex (OR 1.80, 95%CI: 1.11–2.91; $p = 0.017$). Similarly, those who had tested for HIV before were about 80% more willing to test now compared to respondents who had never tested before (OR 1.79, 95%CI: 1.04–3.08; $p = 0.035$).

Following the backward elimination process, the variables 'sex', 'ever had sex', 'knowledge level of HIV/AIDS', and 'educational support' remained significant in the final regression model. Males were now 42% less willing to test now compared to females (AOR 0.58, 95%CI: 0.36–0.93; $p = 0.025$).Those whose education was being supported by their parents were approximately three times more willing to test (AOR 2.51, 95%CI: 1.21–5.22; $p = 0.014$) and those who had ever had sex were at least twice as willing to test for HIV now (AOR 2.38, 95%CI: 1.36–4.15; $p = 0.002$) compared to those whose education was supported by guardians other than parents and those who had never had sex, respectively.

**Table 3.** Logistic regression output for factors influencing willingness to test for HIV now.

| Variable | Crude Odds Ratio (OR) | | Adjusted Odds Ratio (AOR) | |
|---|---|---|---|---|
| | OR (95%CI) | *p*-Value | AOR | *p*-Value |
| Sex | | | | |
| Female | 1 | | 1 | |
| Male | 0.58 (0.38–0.88) | 0.011 | 0.58 (0.36–0.93) | 0.025 |
| Educational support | | | | |
| Guardian | 1 | | 1 | |
| Parents | 2.36 (1.25–4.45) | 0.008 | 2.51 (1.21–5.22) | 0.014 |
| Ever had sex | | | | |
| No | 1 | | 1 | |
| Yes | 1.80 (1.11–2.91) | 0.017 | 2.38 (1.36–4.11) | 0.002 |
| Level of knowledge of HIV/AIDS | | | | |
| Poor | 1 | | 1 | |
| Good | 1.77 (0.90–3.51) | 0.101 | 2.43 (1.13–5.23) | 0.023 |

**Table 3.** *Cont.*

| Variable | Crude Odds Ratio (OR) | | Adjusted Odds Ratio (AOR) | |
|---|---|---|---|---|
| | OR (95%CI) | *p*-Value | AOR | *p*-Value |
| * Age (years) | | | | |
| ≤20 | 1 | | | |
| >20 | 0.71 (0.46–1.10) | 0.126 | | |
| * Hall of residence | | | | |
| Privately-owned | 1 | | | |
| University-owned | 1.26 (0.73–2.19) | 0.401 | | |
| * Marital status of parents | | | | |
| Married | 1 | | | |
| Single/Divorced/Widowed | 1.30 (0.83–2.03) | 0.257 | | |
| * Usual residence outside campus | | | | |
| Rural | 1 | | | |
| Urban | 1.17 (0.72–1.89) | 0.522 | | |
| * Ever tested for HIV | | | | |
| No | 1 | | | |
| Yes | 1.79 (1.04–3.08) | 0.035 | | |
| * School | | | | |
| SAHS | 1 | | | |
| SBBS | 0.62 (0.30–1.27) | 0.192 | | |
| SOM | 0.81 (0.42–1.54) | 0.515 | | |
| SONAM | 0.87 (0.51–1.49) | 0.610 | | |
| SOP | 0.62 (0.25–1.52) | 0.294 | | |
| SOSEM | 0.08 (0.01–0.73) | 0.025 | | |
| * Ethnicity | | | | |
| Akan | 1 | | | |
| Ewe | 1.02 (0.63–1.63) | 0.946 | | |
| [a] Others | 1.04 (0.58–1.85) | 0.900 | | |

[a] Others are defined under Table 2 * Variables without adjusted odds ratios did not appear in the final model following the stepwise regression.

## 4. Discussion

The willingness to test for HIV now, among others, was assessed among university freshmen considering their relative freedom from parental supervision in addition to university campuses being places for risky sexual behaviour that increases the risk of HIV infection [20,31]. About 66% of respondents were willing to test their status and this was mostly driven by curiosity to find out their HIV status.

The present study found nearly two-thirds of respondents being willing to test for HIV and this compares favourably with the 62.7% reported among university students in southern Ghana [12]. It was, however, markedly lower than the 85.7% reported in another group of university students in northern Ghana [16]. It is not clear what specific dynamics contributed to the marked variation seen in the northern Ghana study. They may have had particular exposures that made them nearly homogeneously willing to test for their HIV status. Both studies [12,16] involved other students aside freshmen but this did not appear to make much of a difference as one of them [12] reported findings similar to the present study.

Among those unwilling to check their HIV status now, two clearly 'worrisome' reasons given were that they can never get infected, similar to previous study findings also in university students [32], and that HIV has no cure, so it is not necessary to know one's status. These perceptions seem not to align with the respondents' high level of good knowledge of HIV/AIDS of 90.8% prevalence observed in this study. It may also suggest that having good knowledge of HIV/AIDS may not be enough to counter the respondents' perceptions of the disease. Thinking one can never get infected with HIV may have

religious belief undertones that possibly overshadow scientific knowledge, but this was not investigated. Majority of the respondents were Christians and perhaps religion may have also played a role in why some participants were not willing to get tested. Nevertheless, such high prevalence of good HIV/AIDS knowledge among university students resonates with earlier reports [12,33]. Furthermore, perceptions such as the ones indicated hinder control efforts to reduce HIV transmission and progress towards the 2030 global targets of 95-95-95, i.e., 95% of people living with HIV knowing their status, 95% of those who know their status initiating antiretroviral medication, and 95% of those on medication achieving viral suppression [3]. Again, such apparent ineffectiveness of HIV/AIDS knowledge may also reflect how much the HIV/AIDS awareness campaign in Ghana by the National HIV/AIDS Control Programme (NACP) has dwindled in the past few years possibly due to their current focus on key populations such as female sex workers and their clients as well as men who have sex with men [34]. It is thus pertinent for the Ghana AIDS Commission and NACP to revitalize awareness campaigns focusing on the universality of risk using new mass media platforms such as social media in addition to the traditional ones and the need to check one's status through the promotion of self-testing kits.

Willingness to test lagged behind the near-homogeneous good knowledge of HIV among respondents in the present study. This corroborates previous observations that knowledge of HIV and/or VCT services hardly translates into VCT utilization [12,35,36], although other studies have reported associations between knowledge and VCT uptake [29,30,37,38]. The association between HIV/AIDS knowledge level and willingness to test observed in the study, however, needs cautious interpretation. A key challenge of stepwise regression is that *p*-values can become small compared to other regression approaches [39]. This leads to a high probability of observing exposure variables that have only chance associations with the outcome variable [39]. It is also possible that the observed association arose from the category 'good knowledge' having more than 90% of respondents.

Less than 25% of respondents in the current study had ever tested for HIV compared to about 45–51% reported in other studies in Ghana and Kenya among young people [12–14,40]. These figures are rather low and reflect the general observation of low HIV testing uptake among young people including university students [7,11]. It is possible such young people can be HIV-positive and remain undiagnosed for a long time until overt clinical manifestations raise suspicions. Appreciable transmission can occur within this period and present formidable challenges for control efforts. Similar to previous findings in Ethiopia [11], indifference to knowledge of one's HIV status, stigma, fear of positive results, and a perceived lack of confidentiality among health workers were the key reasons given for not testing for HIV before. A lack of knowledge of where to access VCT services did not appear to be a challenge in the present study as only one respondent gave that reason for not having tested previously. In a study in northern Ghana, 70% of the participating university students had previously tested for HIV but it is unclear what local circumstances, if any, may have precipitated such appreciable uptake of VCT services [16].

More than half of previous HIV tests among the study participants were done at the request of health care providers. Taking a broader perspective, out-patient department hospital attendance can be a vehicle for reaching young people with VCT services where they are offered the opt-out testing approach irrespective of the symptoms they are presenting with. This approach is used at antenatal clinics [41].

In the current study, more females than males had previously tested for their HIV status and the same was observed with respect to willingness to test now. These findings are similar to previous reports in Ethiopia and China [11,15] and may be linked to the general observation that females have overall better health-seeking behaviour than males [42,43]. An alternate explanation may be that females are more vulnerable and may be more easily preyed upon sexually. Five female respondents gave a history of rape as a reason for their willingness to test for HIV at the time of data collection. The finding that more females than males had tested for HIV before or were more willing to test now, however, contrasts with other reports in which more male students had either previously tested or were

more willing to test for HIV [12,13,16,44]. The reason for the variation is unclear but a Kenyan study describes a situation where male students considered it a fun activity going to the VCT centre in groups to check their status [13]. The findings of another study in particular [16] need to be interpreted with caution as male participants were more than twice the number of females to begin with.

Respondents with a previous experience of sex were twice as willing to test for HIV now compared to those with no such history. Sexual intercourse remains the most frequent means of HIV transmission [45] and those who have engaged in sex before may perceive an increased risk of infection. Many young people who have never had sex are less likely to get tested as they tend to think of sex as the only means of transmission [40].

The main source of financial support for education, as a risk factor for willingness to test, is more relevant for female students whose main financial support for education comes from non-parents including philanthropists. Such support may be underlined by male-driven power imbalances that marginalize these females as they provide sexual favours in return for the financial support [46]. It is expected that such students would perceive a higher risk of infection and be more willing to test for HIV. In the current study, however, participants whose main educational support was from their parents were twice as willing to test for HIV compared to those who had support from non-parents. This could be because close to 9 in 10 respondents had their parents supporting their education. There were thus very few in the category of non-parents to make any statistical impact. It could also be interpreted as an expression of respondents' trust in their parents to accept the outcomes of their HIV tests and support them fully to seek health care.

This study had some limitations. It was conducted among freshmen of a single university in Ghana and is thus of limited generalizability. Secondly, the institution is for the training of health professionals only and thus may not reflect first year students of other fields of study. The method used to categorize 'level of knowledge' may be subjective and can limit comparison to other studies; however, this method has been previously used in other studies and referenced above. Except for the variable 'sex' where the adjusted odds ratio in the final multivariate model remained the same as in the bivariate model, the other variables 'education support', 'ever had sex', and 'knowledge level of HIV/AIDS' had their adjusted odds ratios higher than the unadjusted estimates. This may suggest some confounding from the other predictor variables. In addition, this observation may also be due to some missing data in the other variables in the larger multivariate model. Nevertheless, the findings are comparable with those of other studies and provide useful insights regarding knowledge of HIV and attitudes towards uptake of HIV testing services in the specific population of university freshmen. The study used a quantitative approach only but qualitative enquiries in addition would have enabled better exploration of the reasons given for not having ever tested or for being unwilling to test for HIV.

## 5. Conclusions

With about a third of study participants unwilling to test for HIV now and some of the reasons underlying their unwillingness to test suggesting a possible conflict between personal beliefs and knowledge of HIV/AIDS, it may be necessary for school authorities to take steps to fill these gaps. In addition, the NACP needs to step up on large-scale awareness campaigns once again to improve interest in HIV-related matters and to help empower the populace to affect some behaviour change. Further qualitative research will help throw more light on the gender dynamics underpinning willingness to test for HIV.

**Supplementary Materials:** The following supporting information can be downloaded at: https://www.mdpi.com/article/10.3390/venereology1020015/s1, File S1: Data Collection Tool/Questionnaire, File S2: The dataset from which the results presented are derived.

**Author Contributions:** Conceptualization, M.T.-M. and G.D.A.; methodology, M.T.-M., G.D.A., E.K.A. and J.O.; validation, M.T.-M., G.D.A. and E.K.A.; formal analysis, J.O.; investigation, M.T.-M. and G.D.A.; resources, M.T.-M.; data curation, M.T.-M., J.O. and G.D.A.; writing—original draft

preparation, M.T.-M.; writing—review and editing, J.O., G.D.A. and E.K.A.; visualization, J.O., G.D.A., E.K.A. and M.T.-M.; supervision, G.D.A. and J.O.; project administration, M.T.-M. and G.D.A. All authors have read and agreed to the published version of the manuscript.

**Funding:** This research received no external funding.

**Institutional Review Board Statement:** The study was conducted in accordance with the Declaration of Helsinki and approved by the University of Health and Allied Sciences Research Ethics Committee (Protocol Identification Number: UHAS-REC A.12 [150] 20-21; Date of Approval: 12 July 2021).

**Informed Consent Statement:** Written informed consent was obtained from all subjects involved in the study.

**Data Availability Statement:** Data supporting reported results have been provided as part of the Supplementary Materials.

**Acknowledgments:** The authors are grateful to the study participants for their time.

**Conflicts of Interest:** The authors declare no conflict of interest.

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
