# Peer review of "Willingness to Test for Human Immunodeficiency Virus (HIV) Infection among First-Year Students of a Public University in the Volta Region of Ghana"

_venereology, doi:10.3390/venereology1020015_

Round 1

Reviewer 1 Report

Thank you for giving me the opportunity to review the article. The author conducted a study focusing on the willingness to test for HIV infection among first-year university students in Ghana. The topic is socially important, but there are several problems in the manuscript. Therefore, the reviewer thought that the manuscript should be revised before further considerations. I listed my comments below.

Comments:

Abstract:

1.      The authors should add the names of the universities which they conducted this study.

2.      It is difficult to understand “good HIV/AIDS knowledge”. The authors should describe it briefly.

Material and Methods:

3.      How did the authors conduct the test? The authors should describe it in more detail. Did students complete the survey at different times?

4.      The way to determine the cut-off value of knowledge on HIV/AIDS is unclear. The authors should justify the score with related references.

5.      The authors should select the variables which include multivariate analysis using a statistical selection process (e.g., stepwise selection).

Results:

6.      In the Table 3, the authors should show “Yes/No” with a consistent manner. The variable about “Ever had sex” should replace “No/Yes” with “Yes/No”.

Discussion:

7.      The authors should discuss about the limitations of this study in more detail. For example, the authors conducted the study in a campus, and potential readers do not know about the educational level of these students.

8.      What is Ghana's university enrollment rate? This is also important to consider.

9.      The way to assess the knowledge level is also can be a limitation.

Reviewer 2 Report

This introduction introduces an interesting and relevant viewpoint on an important topic. However, the authors did fail to provide a reference for the opening sentence regarding the morbidity and mortality of HIV/AIDS. The flow of the introduction was poorly written and only sometimes provided appropriate references. The major weakness of this introduction is the lack of clear objectives. Although the study mentions the literature gap, there is no goal stated.

This paragraph does not belong to the introduction and pertains to methods. “A cross-sectional study was conducted among first-year students of the University of Health and Allied Sciences in Ghana to ascertain their willingness to test for HIV and influencing factors, knowledge of HIV among the participants and the proportion who had ever tested for HIV. The study is expected to provide useful insights to 81 guide policy makers in promoting HIV counselling and testing among the youth with the hope of 82 early diagnosis and subsequent linkage to treatment.”

The authors failed to present the type of courses the first-year students are exposed to and the link between courses and their willingness to test.

Methods

The authors failed to state this study was a survey. The study failed to state how the survey was constructed and if they used any framework for development. Furthermore, the major concern is that the authors might have used a validated questionnaire that was not refenced.

Another failure of this study stays in this paragraph “A correct response was scored 1 mark. For questions with multiple correct responses a maximum of 2 marks could be attained. Selection of the “I don’t know” option attracted a score of zero. Thus, a total of 10 marks could be obtained in the section of the questionnaire assessing knowledge level of HIV/AIDS and a score of ≤6 was considered poor knowledge while a score of ≥7 was considered good knowledge. The score categorization was arbitrary and chosen to reflect a desired need for people to have adequate knowledge of HIV/AIDS as this aids in prevention of the disease. The questionnaire was pretested among ten second-year Physician Assistantship students and appropriate changes made to its structure before it was used.”

Since the authors did not write this manuscript as attitudes, knowledge, and behaviors where the correct answer must be selected, the entire section must re-address it. The authors should consult on how manuscripts are written to describe a survey. Words such as “scored 1 mark” are more didactic related. Thus, the authors should write “choices were selected” etc.

The language is inappropriate in some sentences. For example, “The study involved only gathering information by means of questionnaires with no health risk to the participants. Participants were not provided with any pecuniary or material benefits.” It would be more appropriate and reader-friendly to use the term “monetary” if that is indeed the point the authors are trying to convey.

Results

One of the major concerns about this section stays in the writing style. For example, the authors should select one style to present their results consistently whether that is percentage or proportions. The authors should be aware on how to write for scientific manuscripts and round up or down the numbers. “Four hundred and twelve (412) first-year students participated in the survey and same number of questionnaires were analysed. The mean age (SD) was 20.04 years (1.65) years. Participants 158 years and below (18-20 years) accounted for 67.2% (274/408) of participants. “

The authors should rephrase this paragraph “Respondents whose education is supported by their parents have more than twice the odds of willingness to test for HIV now compared to those supported by guardians (OR 2.36, 95%CI: 1.25–4.45; p=0.008). Participants who have had sexual intercourse before were 80% more willing to get tested now than those who have never had sex (OR 1.8, 95%CI: 1.11-2.91; p=0.017). Similarly, those who had ever tested for HIV were also about 80% more willing to test now than respondents who had never tested before and this was statistically significant.”

Discussion

Although some parts of the discussion were concise, some sentences need additional information. For example, “Presumably, the respondents may have been ill or predisposed to risky situations like rape and the tests done on account of their presenting complaints. “Are the authors stating that the participants are suffering from HIV? If so, the sentence must be referenced.

 The discussion also compared this study’s results with other similar studies and their results. They analyzed how each of the studies were similar and how they differed which helped to give context to the results. There was a sentence in the discussion that quoted that some respondents “resort to the God factor”. This sentence did not read quite well, and it could offend some readers. Thus, the sentence must be reworded to say that perhaps religion plays a role in why certain participants did not want to get tested would go over better for certain audiences. One strength of this manuscript is the call to action for the Ghana AIDS commission and the NACP as it provided tangible steps that could be done moving forward based on the research.

Another weakness of this manuscript is the lack of limitations. The authors stated, “Qualitative enquiries would have enabled better exploration of the reasons given for not having ever tested or for being unwilling to test for HIV but these were not included in the present study.” Should the reader interpret this sentence as a limitation?

Conclusion

The conclusion was well-written and summarized the manuscript concisely. This conclusion restated the important findings from the study and restated the main action steps for organizations to take to address this issue. It also provided other investigators with interest in this topic an idea to do a project around qualitative research so that is positive to advance the future of this topic.

Round 2

Reviewer 2 Report

The authors addressed my comments